# Branch-Train-MiX:
# Mixing Expert LLMs into a Mixture-of-Experts LLM

**Sainbayar Sukhbaatar    Olga Golovneva    Vasu Sharma    Hu Xu    Xi Victoria Lin**
**Baptiste Rozière    Jacob Kahn    Daniel Li    Wen-tau Yih    Jason Weston    Xian Li**

FAIR at Meta
{sainbar,xianl}@meta.com

## Abstract

We investigate efficient methods for training Large Language Models (LLMs) to possess capabilities in multiple specialized domains, such as coding, math reasoning and world knowledge. Our method, named Branch-Train-MiX (BTX), starts from a seed model, which is branched to train experts in embarrassingly parallel fashion with high throughput and reduced communication cost. After individual experts are asynchronously trained, BTX brings together their feedforward parameters as experts in Mixture-of-Expert (MoE) layers and averages the remaining parameters, followed by an MoE-finetuning stage to learn token-level routing. BTX generalizes two special cases, the Branch-Train-Merge method, which does not have the MoE finetuning stage to learn routing, and sparse upcycling, which omits the stage of training experts asynchronously. Compared to alternative approaches, BTX achieves the best accuracy-efficiency tradeoff.

## 1   Introduction

In recent years, Large Language Models (LLMs) have shown impressive performance in a wide-range of tasks (Brown et al., 2020; Touvron et al., 2023; Achiam et al., 2023), including code generation (Li et al., 2022b; Rozière et al., 2023), solving math problems (Azerbayev et al., 2023), multilinguality (Zhao et al., 2024), etc. Training such LLMs requires a large amount of compute and data, exceeding thousands of GPUs and trillions of tokens. The training parallelization is typically done by maintaining multiple copies of the model on different GPUs and keeping them synchronized after each weight update. The cost of this frequent communication is the main bottleneck in scaling the training to more GPUs. Besides this issue, synchronized training is more vulnerable to hardware failures as a single failed GPU can cause the whole training to halt (Zhang et al., 2022; Gemini Team, 2023).

Recent work by Li et al. (2022a) proposed the Branch-Train-Merge (BTM) method for embarrassingly parallel training of LLMs without any synchronization for improving the throughput of pretraining. It starts by creating multiple copies of a seed LLM, then separately training each copy on different subsets of data. This results in multiple independent LLMs that do not share any parameters and each LLM is an expert specializing in its own data distribution, such as knowledge domains, languages or even modalities. At test time, an input prompt is classified into one or more of the domains, and then the final outputs are formed from the corresponding expert models which are combined to predict the next token. While this approach makes training more efficient, its main drawback is the lack of a unified single model making it impossible to do further supervised finetuning (SFT) or reinforcement learning from human feedback (RLHF) finetuning (Ouyang et al., 2022), both of which can boost performance further, and are crucial steps in building aligned LLMs.

A separate line of work for reducing the computational footprint of LLMs is the Mixture-of-Experts (MoE) approach (Jacobs et al., 1991; Shazeer et al., 2017), where only a subset of parameteters are active at any given time. In particular, MoE is applied to the feedforward sublayer of Transformers (Fedus et al., 2022; Roller et al., 2021; Lewis et al., 2021), allowing

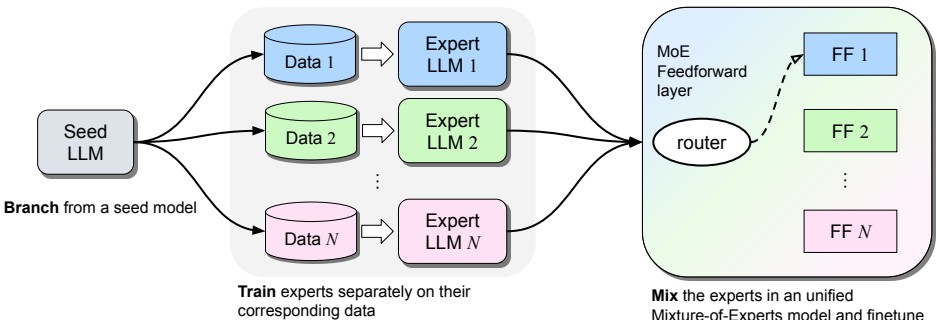

Figure 1: **The Branch-Train-MiX (BTX) method** has three steps: **1) branch** from a pretrained seed LLM by making multiple copies of it; **2) train** those copies separately on different subsets of data to obtain expert LLMs; **3) mix** those expert LLMs by combining them into a single LLM using mixture-of-experts feedforward (FF) layers, and finetuning the overall unified model.

the total number of parameters to grow without additional computation. LLMs scaled in this way have shown impressive performance on downstream tasks (Jiang et al., 2024; Xue et al., 2024). Unlike Branch-Train-Merge, Mixture-of-Experts are often trained in a fully synchronized fashion, and the communication cost increases with the number of experts due to all-to-all communication.

In this paper, we aim for the best of both worlds, combining the advantages of Branch-Train-Merge and Mixture-of-Experts, while mitigating their disadvantages. We achieve this by training multiple expert LLMs separately as in the Branch-Train-Merge method, but subsequently combine those experts into a single model using an MoE architecture. More specifically, the feedforward sublayers from all the expert LLMs are brought together into a single MoE module at each layer, and a router network selects which feedforward expert to use at every token. We merge other modules of the expert LLMs, including self-attention layers, by simply averaging their weights. Then the resulting model is MoE-finetuned on all the combined data by continuing training, so that the router can learn to mix the expert feedforward (FF) modules. Figure 1 shows an overview of this method, which we call *Branch-Train-MiX* (BTX).

The main advantage of BTX compared to MoE is that expert training is embarrassingly parallel and asynchronous, reducing communication cost and increasing training throughput. Compared to Branch-Train-Merge, the final BTX model is a unified neural network that can be finetuned or used like any other standard LLM. The final BTX model will not significantly increase inference FLOPs compared to the seed model since it is sparsely activated, despite having a much larger number of parameters.

We conduct our experiments using LLAMA-2 7B (Touvron et al., 2023) as a seed model and train expert LLMs on different subsets of data corresponding to the domains of math, code and Wikipedia. With the original LLAMA-2 7B weights added as a fourth expert, we finetune the combined MoE model for a relatively short period compared to the pretraining process. The resulting BTX model brings significant improvements over the seed model on tasks across various domains, especially bridging the gap with specialized models on math and code related tasks, while retaining performance on the original capabilities where specialized models suffer from catastrophic forgetting. BTX outperforms BTM on all tasks demonstrating the benefits of learnt routing through MoE finetuning. Compared to purely MoE training such as sparse upcycling, BTX is more compute efficient with higher training throughput and more balanced performance across tasks in different domains.

## 2 Related Work

**Asynchronous parallel training** Reducing communication between training workers for computational efficiency is a major topic of study for training deep learning systems. Zhang et al. (2015) introduced a method that allows model instances on different workers to diverge

from each other, thus eliminating the constant need of synchronization. Instead, the workers are loosely synchronized to master weights using elastic averaging from time to time. A more recent work by Douillard et al. (2023) showed that less frequent synchronization of diverged workers by averaging their weight changes and applying Nesterov momentum works well in practice for training LLMs. The Branch-Train-Merge method (Li et al., 2022a; Gururangan et al., 2023) takes parallel training to the extreme by running multiple training processes completely independently. Each training process uses specific domain data, thus the corresponding model becomes an expert in that domain. Finally, the output distributions of those expert models are averaged to make a next token prediction. Which experts to average is decided by classifying the input into one or more of the domains. Wortsman et al. (2022) showed simply averaging parameters of separately trained models improves performance, but the models only differed in their hyperparameters.

**Mixture-of-Experts** MoE is used to scale deep networks in Shazeer et al. (2017) using a simple Top-K routing scheme. Since the routing decisions are discrete and thus cannot be trained by gradient descent, various training methods have been explored for the Transformer architecture (Fedus et al., 2022; Lewis et al., 2021). Surprisingly Roller et al. (2021) showed that even a fixed routing scheme without any learning works well, if the routing is done via a random mapping based on input tokens. In larger scale experiments with recent LLMs, Jiang et al. (2024) demonstrated that the MoE approach can match the performance of dense LLM counterparts using a much smaller number of active parameters. A study by Dai et al. (2024) showed the advantage of more fine-grained experts, as well as having a shared expert that always stay active. More similar to our work, Gururangan et al. (2021) makes experts in feedforward layers specialize to specific domains using a domain-conditioned fixed routing, but it lacks the asynchronous training of our approach.

**Continual learning** Our method relates to continual learning (Awasthi & Sarawagi, 2019) because domain experts are trained on datasets with different distributions from the initial data used for training the seed model, which is implemented by continued training after branching. Specifically, our approach is related to parameter isolation methods (Lange et al., 2019) as we have different parameters for different domains. Aljundi et al. (2016) also creates a new copy of a model to train on each domain. Rusu et al. (2016) adds a new model with a new domain, but connects it to the previous models so the previously learned features can be used. Rozière et al. (2023) showed continual training of a seed LLM on a specific domain of code can produce a strong domain expert model, and this converges much faster than starting from scratch. For training a math expert, starting from a code expert rather than a general LLM was shown to be more beneficial (Shao et al., 2024; Azerbayev et al., 2023).

## 3 Branch-Train-MiX

Given an existing LLM $\mathcal{M}$ which has been pretrained on a large corpora covering a wide variety of topics, we aim to improve its performance on $N$ areas of expertise. This is achieved by continued pretraining with corresponding training datasets $\mathcal{D} := \{D_1, \ldots, D_N\}$, each related to a specific knowledge domain such as math, code, etc. The proposed method contains three stages: Branch, Train, and MiX.

### 3.1 Branch & Train: Embarrassingly Parallel Expert Training

Initializing from the seed model $\mathcal{M}$, we train $N$ expert LLMs $\{\mathcal{M}_1, \ldots, \mathcal{M}_N\}$, with each model $\mathcal{M}_i$ being trained on the corresponding dataset $D_i$ in the same manner as during pretraining, using the usual LM objective. Since each expert model $\mathcal{M}_i$ can be trained in complete separation from the others, the whole training process becomes $N$-way embarrassingly parallel. This training paradigm has several benefits in large-scale distributed training. It allows linear scaling of overall training throughput when scaling up the size of compute, while joint training often faces uncertain performance from increasing batch size. It has lower all-to-all communication cost. It is also more resilient, as a single training failure will only affect one of the $N$ training processes instead of halting the entire training.

After all the expert training is finished, we will end up with $N$ different LLMs, with each specializing in a specific distribution. At this point, the Branch-Train-Merge method (Li et al.,

2022a; Gururangan et al., 2023) uses these domain experts as is, choosing which expert to use by determining which domain the input belongs to at inference time. Usually multiple experts are chosen, and their final output distributions are simply averaged to generate the next token. Our BTX approach, in contrast, merges these domain experts back into a single LLM that is finetuned further, as we will describe in the next section.

## 3.2 MiX: Combining Separate Experts to be a Mixture-of-Experts

We employ a Mixture-of-Experts approach to combine the domain expert models $\mathcal{M}_i$. However, instead of using the classical procedure of mixing the final outputs from $\mathcal{M}_i$, we do a more fine-grained mixing by performing MoE within each layer of a Transformer. In particular, we combine the different feedforward sublayers from the domain experts into a single MoE sublayer. If $\text{FF}_i^l(x)$ is the feedforward sublayer at the $l$-th layer of the $i$-th domain expert $\mathcal{M}_i$, then the combined MoE layer for input representation $x$ at layer $l$ will compute:

$$\text{FF}_{\text{MoE}}^l(x) = \sum_{i=1}^{N} g_i(W_l x) \text{FF}_i^l(x).$$

Here $W_l$ is a linear transformation and $g$ is a routing function, which usually has sparse output and hence switches on only some experts. Since we can skip computing $\text{FF}_i^l(x)$ if the corresponding router output is zero, the actual computation of $\text{FF}_{\text{MoE}}^l(x)$ will be much more efficient than computing all domain experts. However, routing decisions can change from token to token, so one input sequence can employ all the domain expert FF layers if needed, even when only a few are accessed at any given token. In our experiments, we use Top-k (k=2) routing where $g(W_l x) = \text{SoftMax}(\text{TopK}(W_l x))$, unless otherwise stated.

For the self-attention sublayers, we combine the different domain experts by simply averaging their weights. The motivation behind this is the assumption that the self-attention layers are less domain specialized than the feedforward layers. We do the same averaging for the remaining parameters (embeddings, etc.) as well.

Note that the only new parameters we introduce are the router's transformation parameters $W_l$, which are negligible in size compared to the rest of the network. Nevertheless, those new parameters need to be finetuned, so the router can make optimal decisions in selecting which domain $\text{FF}_i$ to use. In addition, funetuning is helpful because the self-attention weights are constructed by averaging, and are likely not optimal. Overall, the entire system has not been optimized for working together at all in the embarrassingly parallel training framework, but our hypothesis is that even a small amount of combined finetuning might make large improvements.

## 3.3 Variations

We also experimented with several variations of our method.

**Load balancing** A common problem with MoE is the emergence of dead experts, which do not get activated by the router at all. Common routing methods like Top-k are unlikely to escape from such a situation because a dead expert is never in the top-k selection, and therefore never receives a training signal. Load balancing offers a simple solution by adding an extra loss term that encourages the experts to be utilized equally. We use a loss term similar to (Fedus et al., 2022):

$$\mathcal{L}_{\text{LB}} = \alpha N \sum_{i=1}^{N} u_i p_i \quad \text{where } u_i = \frac{1}{|\mathcal{B}|} \sum_{x \in \mathcal{B}} g_i(W_l x) \text{ and } p_i = \frac{1}{|\mathcal{B}|} \sum_{x \in \mathcal{B}} \text{SoftMax}_i(W_l x).$$

Here $\mathcal{B}$ is the current data batch, and $\alpha$ is a hyperparameter. This loss is computed in each layer and added to the NLL loss.

**Routing method** Besides Top-k routing, we also experiment with other routing methods:

- **Switch:** It is a Top-1 routing method proposed by Fedus et al. (2022).
- **Soft routing:** We use softmax as the routing function $g$, so all experts are activated both during training and inference. While it is likely to provide the best performance, it comes at the expense of increased compute.

|  | Math | | Code | | General knowledge | | |
|---|---|---|---|---|---|---|---|
|  | **GSM8K** | **MATH** | **HumanEval** | **MBPP** | **NQ** | **TriviaQA** | **MMLU** |
| LLAMA-2 7B | 14.7 | 2.5 | 12.8 | 20.8 | 16.4 | **58.5** | 46.1 |
| Math expert | **39.5** | **18.8** | 25.0 | 33.6 | 14.4 | 37.1 | **52.0** |
| Code expert | 12.0 | 4.0 | **31.7** | **40.2** | 11.5 | 29.9 | 39.6 |
| Wikipedia expert | 11.7 | 3.1 | 11.0 | 15.2 | **21.8** | 57.2 | 43.1 |
| BTX, Top-2 | 37.1 | 17.8 | 28.7 | 39.4 | 24.8 | 57.1 | 52.5 |

Table 1: Individual domain expert LLM performance on representative tasks, compared to the seed model LLAMA-2 7B. As expected, the code and math experts excel at their corresponding domain tasks. The Wikipedia expert performs better on Natural Questions (NQ), but the math expert has the best score on MMLU (MMLU contains many math subjects and math training is shown to help on this task (Shao et al., 2024)).

- **Sample Top-1:** We use the gumbel softmax (Jang et al., 2016) for $g$. At training time, we generate a soft sample from the gumbel softmax, but zero out all its values except the largest one. Then we compute only one expert corresponding to this largest value, omitting the other expert computations. At inference time, we simply do hard sampling. We anneal the temperature to a sharp distribution at the end of training to gradually reduce the discrepancy between training and inference.

## 4   Experiments

### 4.1   Experimental Setup

We base our experiments on the setup used for LLAMA-2 pretraining (Touvron et al., 2023). In particular, we use the LLAMA-2 7B model as our seed model.

#### 4.1.1   BTX Training

We use the pretrained Llama-2 (Touvron et al., 2023) with 7B parameters as our seed model. After making three copies of the seed model LLAMA-2 7B, we continue training them on the following domain datasets to derive three domain experts:

- **Math:** The same data sources and mixture used in Llemma (Azerbayev et al., 2023) model training. To be comparable to Llemma, we train on the same amount of data as well, i.e. 48k steps with 201B tokens in total.

- **Code:** The same data sources and mixture of code data used in CODELLAMA pretraining (Rozière et al., 2023). The code expert LLM is trained for 50k steps with 210B tokens in total to be comparable with the math expert.

- **Wikipedia:** Wikipedia documents extracted between June to August 2022. The data was preprocessed to remove hyperlinks, comments and other formatting boilerplate. Since this is a smaller dataset, we train on a total of 42B tokens.

While we can proceed with only these three domain experts, we also include the original seed LLM as a "generalist" expert so that its general knowledge is transferred to the final model. Thus we mix these four expert models into a single MoE model as described in Section 3.2. Then we finetune this MoE model on all the data sources used to train the four experts (including the original LLAMA-2 7B pretraining data for the generalist expert) and train for another 80B tokens. The detailed sampling ratio of datasets as well as other training details are described in Appendix A.

#### 4.1.2   Baselines

We compare to the following baselines:

- **LLAMA-2:** We compare to the original LLAMA-2 7B that we use as a seed model, as well as LLAMA-2 13B.

|                                 | Math | Code | Knowledge | Reasoning | MMLU | Average |
|---------------------------------|------|------|-----------|-----------|------|---------|
| *Specialized LLMs*              |      |      |           |           |      |         |
| CODELLAMA 7B                     | 8.1  | 36.3 | 22.2      | 56.6      | 38.6 | 37.9    |
| LLEMMA 7B                        | 28.0 | 33.5 | 17.2      | 38.8      | 33.5 | 32.1    |
| *Generalist LLMs*               |      |      |           |           |      |         |
| LLAMA-2 7B                       | 8.6  | 16.8 | 37.4      | 63.3      | 46.1 | 40.7    |
| LLAMA-2 13B                      | 16.3 | 24.5 | 40.0      | **66.1**  | 52.8 | 45.4    |
| Dense (DM)                       | 18.3 | 25.8 | 39.6      | 63.3      | 49.8 | 44.5    |
| Sparse upcycling (DM), Top-2     | **28.1** | 34.7 | 34.0  | 62.3      | 51.1 | 46.3    |
| BTM, Top-1                       | 21.3 | 36.4 | 26.5      | 61.0      | 44.3 | 43.1    |
| BTM, Top-2                       | 21.5 | **36.6** | 26.9  | 61.2      | 44.3 | 43.4    |
| BTX, Sample Top-1                | 26.4 | 31.5 | 40.1      | 63.7      | **53.2** | 47.3 |
| BTX, Top-2                       | 27.4 | 34.0 | **41.0**  | 63.5      | 52.5 | **47.9** |

Table 2: Test performance of BTX compared to various baselines, including both generalist and specialized pretrained models, on various capabilities aggregated across popular benchmarks. Dense, sparse upcycling, BTM and BTX are trained on exactly the same amount and mixture of data with the exception that BTM does not have the finetuning stage.

- **Dense:** Instead of training separate LLMs on different domain datasets, the dense baseline continues to train the seed LLM with all the data. We use exactly the same training data as BTX, first training on the new domain-specific data used in the experts training stage, followed by the same data mixture that includes the LLAMA-2 pretraining data in the MoE finetuning stage. We call this comparison *data-matching* (DM).

- **Sparse upcycling:** This baseline (Komatsuzaki et al., 2022) initializes a MoE model from the seed model by making 4 identical copies of the feedforward module as experts. We use the Top-2 router with randomly initialized $W_i$ parameters. In addition to training a data matching (DM) baseline with the same data as is used in BTX and the dense baseline, we also train a sparse upcycling baseline with the same amount of GPU-days, i.e. compute-matching (CM), using the MoE finetuning data mixture throughout training. This is equivalent to a special case of BTX which does not contain embarrassingly parallel expert training.

- **Branch-Train-Merge (BTM):** This baseline (Li et al., 2022a) uses the same expert LLMs as BTX (including the original seed model) but uses them directly without building a MoE model. For a given context (input), it selects Top-k expert LLMs based on the similarity between the context and experts' training data. Following the efficient inference method used in Gururangan et al. (2023), both context and experts' training data are embedded via tf-idf. Top-k experts are selected based on cosine similarity to the mean tf-idf embedding of each expert.

- **CodeLlama 7B:** A language model specializing in code (Rozière et al., 2023) by continued training of the same seed model LLAMA-2 7B on code data. It also has other features such as long-context and infilling.

- **Llemma 7B:** A language model specializing in mathematics (Azerbayev et al., 2023) by continued training of CodeLlama 7B on math data.

### 4.1.3   Evaluation

For evaluation, we use the zero- and few-shot performance on multiple benchmarks that test different skills: GSM8K and MATH for math reasoning; HumanEval and MBPP for code generation; Natural Questions and TriviaQA for world knowledge; ARC-Easy, ARC-Challenge, SIQA , PIQA and WinoGrande for reasoning; and MMLU which covers multiple domains. See Appendix B for more evaluation details.

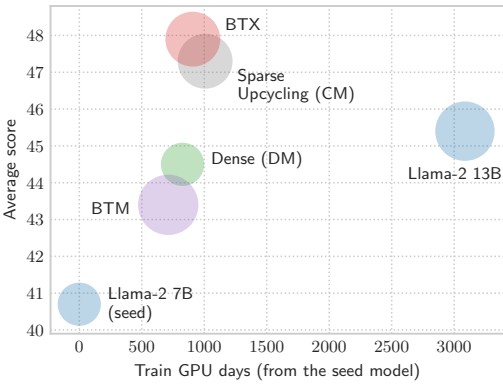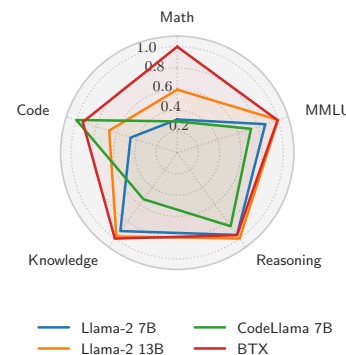

Figure 2: **Left:** The average performance vs training budget of BTX compared to various baselines, with different active parameters at inference time indicated by circle size. All the models except LLAMA-2 13B are trained starting from LLAMA-2 7B using the datasets described in Section 4.1.1. The X-axis shows the total training compute starting from the seed model measured in GPU days[2], and the Y-axis is the average score over all the tasks (as computed in Table 2). The BTX models outperform the baselines that started from the same seed model, as well as LLAMA-2 13B. **Right:** The normalized performance over different domains where the scores are divided by the highest one. We see large improvements for BTX in code (which matches the specialized model) and math tasks compared to the seed model LLAMA-2 7B, even outperforming the LLAMA-2 13B model.

## 4.2 Main Results

### 4.2.1 Overall Performance

**Domain experts excel at their respective tasks.** We first analyze how expert LLMs specialize to specific domains. Results are summarized in Table 1. As expected, individual expert LLMs achieve the best performance in their respective domain, where the math and code domains see especially large improvements. In addition, there are several interesting observations. We see that the math expert training improved its code performance as well, indicating a close relation of these domains. However, such single-domain continued training also suffers from catastrophic forgetting with significant performance drops on some tasks in other domains. For example, the math and code expert are much worse on TriviaQA than the seed model.

**BTX improves all tasks where experts specialize.** Table 2 and Figure 2 (right) show aggregated performance across multiple domains. More detailed per-task results are reported in Table 5 in the Appendix. Compared to the seed model LLAMA-2 7B, BTX models (both Sample Top-1 and Top-2 corresponding to different number of active parameters) improve on all expert domains, such as math, coding and world knowledge without regressing on other tasks such as commonsense reasoning. BTX with Top-2 experts (our default) also approaches the best performance of the specialized models LLEMMA 7B and CODELLAMA 7B in the math and coding domains, while drastically improving over those models on domains that are not their speciality such as world knowledge and commonsense reasoning. Compared to alternative data-matching (DM) methods for continued pretraining such as dense and sparse upcycling, BTX achieves better performance on average with small gaps in the math and coding domains. BTX outperforms BTM by a large margin on average, indicating that MoE finetuning to learn token-level routing is beneficial. Overall, the results demonstrate that BTX is a better performing method for continued pretraining which is

---

[2]The GPU days of Llama-2 13B is an approximate measurement, calculated by doubling the training compute of a 7B model trained with the same amount of pretraining data (according to Touvron et al. (2023) Table 2). Since Llama-2 13B is not trained from the seed model, we simply report their difference in GPU days.

| Routing method | Active parameters (B) | | Average score | | |
|---|---|---|---|---|---|
| | Training | Inference | train=10B | train=40B | train=80B |
| Switch Top-1 | 6.7 | 6.7 | 24.7 | - | - |
| Soft routing | 19.7 | 19.7 | 35.8 | 37.3 | - |
| Sample Top-1 | 6.7 | 6.7 | 33.0 | 35.3 | 36.9[3] |
| Top-2 | 11.1 | 11.1 | 34.6 | 35.9 | 37.3 |

Table 3: Ablations on different routing methods during BTX training for increasing amount of train tokens. Average score is based on performance on representative tasks including GSM8K, HumanEval, Natural Questions, ARC Challenge and MMLU.

robust to task interference from multi-task learning. BTX also outperforms LLAMA-2 13B on all tasks except reasoning, even though LLAMA-2 13B uses significantly more training compute and has slightly more active parameters.

We further compare BTX with the sparse upcycling baseline in the compute-matching (CM) scenario. Both train on the same data mixture during the MoE stage, but differ in terms of the percent of compute spent on MoE training. While sparse cycling performs close behind BTX, the parallel training of experts increases the training throughput of BTX, as is shown in Table 6 in the appendix. As a result, BTX can train with more than $2\times$ the data than pure MoE given the same training compute budget, and achieves slightly higher average performance across all domains.

In Table 1, we directly compare BTX with the original expert LLMs. While we see some degradation in the math and code domains, the final BTX model has the best overall performance and competitive with the expert LLMs on their respective domain tasks.

### 4.2.2 Better compute-performance tradeoff

We compare BTX with baselines in terms of compute efficiency in Figure 2 (left). The X-axis shows the total training compute starting from the seed model measured in GPU days, which includes the domain expert training and finetuning of the MoE model. The Y-axis measures the overall performance reported in Table 2.

**Better performance than dense and BTM.** Despite that the MoE training stage uses a fraction of the total training budget in pretraining (for example, LLAMA-2 pretraining uses 2T tokens), BTX brings steep improvements on general capabilities compared to alternative continued pretraining approaches such as multi-task learning of the dense model and Branch-Train-Merge.

**More efficient than sparse upcycling.** As a special case of BTX, sparse upcycling without expert training outperforms dense and BTM but not BTX, given the same or larger compute budget. The compute efficiency gains of BTX are from the embarrassingly parallel training of experts before MoE finetuning.

In terms of the active number of parameters (shown as circle sizes in 2 (left)), the MoE models are similar to the LLAMA-2 13B model. BTX uses less than half of the additional training compute compared to LLAMA-2 13B, but demonstrates improved performance on expert domains (math, code, and knowledge) and achieves better overall performance. This indicates that BTX's training is more effective for the late stage of pretraining than using the same training protocol throughout the entire of pretraining.

### 4.3 Ablations & Analysis

### 4.3.1 Ablations of BTX training

First, we compare the different routing methods with varying amount of active parameters for different amounts of finetuning. For fair comparison, load balancing is not used in any

---

[3]Sample Top-1 training is more efficient, so we trained for 160B tokens for the same compute.

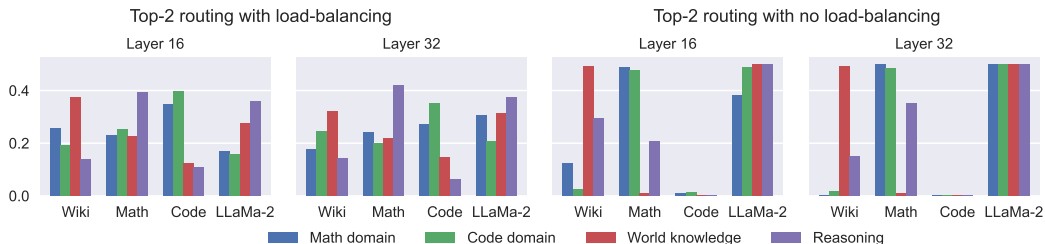

Figure 3: BTX token routing decisions at middle (16) and top (32) layers to different experts (Wiki, Math, Code, LLAMA-2 7B) for different downstream tasks. The tasks are aggregated by domain: Code (Human Eval, MBPP), Math (GSM8K, MATH), World knowledge (Natural Questions, TriviaQA), and Reasoning (ARC-Easy, ARC-Challenge, SIQA, PIQA, and WinoGrande). We observe that Top-2 routing with load balancing (left) ensures a more uniform distribution of the load between experts compared to Top-2 without load balancing (right).

of them. Results are shown in Table 3. For Switch routing, we set its capacity factor to 1.5 (a hard limit after which routed tokens will be dropped). We found the Switch router to be subpar in average performance. The soft routing performs the best, but that is expected since it lacks sparsity and has the highest number of active parameters. Overall, the Top-2 routing gives us a good balance between performance and efficiency.

We also ablate additional design choices of BTX, with results summarized in Table 7 in the appendix. We found that MoE training without load balancing performs worse on the coding task (HumanEval), but has higher math (GSM8k) accuracy. The routing analysis in the next section will give more insight into this trade-off. Next, freezing the feedforward modules initialized from each expert, and only training the rest of the MoE model has little impact on performance across all tasks. This suggests that individual experts already gained sufficient domain knowledge during the branch-train stage, while the mix (MoE finetuning) stage mainly trains the other parameters such as averaged weights in the self-attention and the router transformations $W_i$.

### 4.3.2 Routing Analysis

To gain an in-depth understanding of the performance of BTX, we run model evaluations on downstream tasks and examine the routing decisions among the experts. The results are summarized in Figure 3, and we also report detailed ablation results for different BTX setups in Appendix C. Compared to other routing methods, Top-2 routing with load balancing ensures a more uniform distribution of the load between experts. Analyzing the token probability distributions, we observe a shift towards low probability scores across all experts with load balancing, especially closer to the final layers of the model, which contributes to the fair routing. Interestingly, all models without load balance heavily rely on the Math expert, with a low overall contribution from other experts, especially the Code expert. A dead Code expert comes "back to life" with load balancing introduced in training. In fact, it not only becomes visible, but becomes the dominant expert in the math and code domains.

Examples of the routing decisions for Top-2 with load balancing can be found in Table 8 in the appendix. Overall across math domain tasks, tokens are often routed to the Code and LLAMA-2 7B experts. If we look at a more detailed token distribution (Appendix C, Figure 6), we find that the GSM8K task prefers Code and LLAMA-2 experts, while the MATH task relies more on the in-domain Math expert. We hypothesise that this happens because the GSM8K dataset consists of grade school math problems that require common sense knowledge and basic arithmetic operations. Both the Code and World knowledge tasks mostly route to the in-domain Code and Wikipedia experts respectively. As observed earlier in Section 4.3.1, when load balancing is introduced, there are improvements in coding tasks but degradation in math tasks, which can be explained with these changes in domain expert routing. The reasoning tasks in contrast exhibit similar behaviour, and rely equally on Math and generalist LLM's expertise.

## 5 Conclusion

We introduced Branch-Train-MiX (BTX), a simple continued pretraining method to improve an LLM's capabilities. It trains multiple copies of a seed LLM to specialize in multiple domains in an asynchronous and parallel fashion and later merges them back into a single Mixture-of-Experts (MoE) model via finetuning. While the initial parallel training stage brings higher training throughput and scalability, the second MoE finetuning stage makes the final LLM more performant. Our experiments suggest that a generalist LLM's performance can be boosted by continued training on datasets with specialized knowledge and skills using our method. We find that the BTX approach is more compute efficient than training a larger generalist LLM or several separately specialized LLMs. These insights can inform how to allocate compute in late pretraining to achieve a strong generalist model.

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

## A  Training Details

Table 4 shows the exact data mixture ratios used in training each domain expert. For finetuning the MoE model, we sample datasets that used to train math expert, code expert, wikipedia expert and the original LLAMA-2 7B with probabilities 30.16%, 40.31%, 10.30% and 19.23%.

| Domain | Dataset | Sampling ratio (%) |
|---|---|---|
| Math | AlgebraicStack | 13.57 |
| | OpenWebMath | 54.27 |
| | Arxiv | 27.14 |
| | Github | 2.99 |
| | Commoncrawl | 5.01 |
| Code | Code | 82.18 |
| | Natural language related to code | 9.90 |
| | Natural language | 6.93 |
| Wikipedia | Wikipedia | 90.91 |
| | Commoncrawl | 9.09 |

Table 4: Data sources and weights for domain experts.

For BTX with default Top-2 routing, we use load balancing with $\alpha = 0.01$, unless otherwise stated. For the Sample Top-1 routing, we use the temperature annealing schedule $\tau = \max(0.5, -rt)$ from Jang et al. (2016) with $r = 1e - 4$ where $t$ is the number of training steps. For the first layer only, we used soft-routing instead of Top-1.

We use the same optimization hyperparameters for training of the baselines, expert models and MoE models. We use the AdamW optimizer with weight decay 0.1, and anneal the learning rate to the peak of $1e - 4$ with 100 steps of warmup, and decay to 10% of the peak with a cosine schedule. We use a batch size of 4M tokens with a sequence length of 4096.

## B  Evaluation

We use the following tasks for evaluation:

- Math: we report the average performance on GSM8K (8 shot) (Cobbe et al., 2021) and MATH (4 shot) (Hendrycks et al., 2021b) for math reasoning.

- Code: we report the average performance of HumanEval (0 shot) (Chen et al., 2021) and MBPP (3 shot) (Austin et al., 2021) for code generation.

- World knowledge: we report the average performance of Natural Questions (5 shot)(Kwiatkowski et al., 2019) and TriviaQA (5 shot) (Joshi et al., 2017).

- Reasoning: we report the average 0-shot performance of ARC-Easy and ARC-Challenge (Clark et al., 2018), SIQA (Sap et al., 2019), PIQA (Bisk et al., 2020) and WinoGrande (Sakaguchi et al., 2021).

- General: we report performance on MMLU (5 shot) (Hendrycks et al., 2021a) which covers multiple domains.

We use the same evaluation metrics as is used in Touvron et al. (2023) and Rozière et al. (2023): for code tasks (HumanEval and MBPP) we report pass@1, for math tasks (GSM8k and MATH) and knowledge tasks (Natural Questions and TriviaQA) we report exact match, we report accuracy for MMLU and ARC. We use greedy decoding for all generations. Detailed results on all tasks are reported in Table 5.

|  | GSM8K | MATH | Human Eval | MBPP | NQ | Trivia QA | ARC-e | ARC-c | Wino | SIQA | PIQA | MMLU |
|---|---|---|---|---|---|---|---|---|---|---|---|---|
| *Specialized LLMs* | | | | | | | | | | | | |
| CODELLAMA 7B | 13.0 | 3.3 | 31.1 | 41.4 | 11.5 | 32.8 | 67.4 | 34.0 | 62.7 | 46.1 | 72.9 | 38.6 |
| LLEMMA 7B | 39.3 | 16.7 | 25.6 | 41.4 | 9.4 | 24.9 | 28.7 | 26.8 | 50.1 | 37.3 | 51.0 | 33.5 |
| *Generalist LLMs* | | | | | | | | | | | | |
| LLAMA-2 7B | 14.7 | 2.5 | 12.8 | 20.8 | 16.4 | 58.5 | 76.4 | 43.8 | 69.2 | 48.3 | 78.8 | 46.1 |
| LLAMA-2 13B | 28.7 | 3.9 | 18.3 | 30.6 | 16.1 | 63.8 | 77.3 | 49.4 | 73.0 | 50.1 | 80.8 | 52.8 |
| Dense (DM) | 26.7 | 9.9 | 20.7 | 30.8 | 24.0 | 55.3 | 76.7 | 44.5 | 68.9 | 48.3 | 78.2 | 49.8 |
| SU (DM), Top-2 | 37.3 | 18.9 | 29.3 | 40.2 | 18.8 | 49.2 | 76.3 | 43.4 | 66.4 | 47.3 | 77.9 | 51.1 |
| SU (CM), Top-2 | 40.1 | 16.2 | 26.2 | 35.2 | 24.5 | 58.2 | 75.6 | 44.7 | 69.1 | 47.1 | 78.0 | 52.1 |
| BTM, Top-1 | 27.4 | 15.2 | 30.8 | 41.9 | 15.0 | 38.0 | 72.8 | 38.1 | 68.4 | 47.8 | 77.9 | 44.3 |
| BTM, Top-2 | 27.7 | 15.3 | 30.6 | 42.6 | 15.3 | 38.5 | 73.1 | 38.5 | 68.3 | 48.0 | 78.1 | 44.3 |
| BTX, sample Top-1 | 36.9 | 15.8 | 25.6 | 37.4 | 23.7 | 56.4 | 76.7 | 45.0 | 70.6 | 48.0 | 78.2 | 53.2 |
| BTX, Top-2 | 37.1 | 17.8 | 28.7 | 39.4 | 24.8 | 57.1 | 76.9 | 45.6 | 67.9 | 48.7 | 78.7 | 52.5 |

Table 5: Individual task performance of BTX and baselines. SU = Sparse upcycling

|  | MoE compute ratio | Training time (days) | Total compute (GPU-days) | num. tokens (B) | Math | Code | Knowledge | Reasoning | MMLU | Avg. |
|---|---|---|---|---|---|---|---|---|---|---|
| BTX | 23% | 7.8 | 926.1 | 533 | 27.4 | 34.0 | 41.0 | 63.5 | 52.5 | 47.9 |
| SU (CM) | 100% | 7.9 | 1007.1 | 252 | 28.2 | 30.7 | 41.3 | 62.9 | 52.1 | 47.3 |

Table 6: Comparison between BTX and Sparse upcycling (SU) with compute-matching (CM), which is a special case of BTX without the expert training stage as is shown by the first column that 100% of compute is spent on MoE training. We also report total training time, compute and number of training tokens. Comparing both performance on individual domains as well as the average, we can see that BTX has more balanced performance, in addition to higher throughput.

## C  Routing analysis

Layer-by-layer comparison of the routing decision for different router designs and downstream tasks aggregated by task domain is shown in Figure 4. Routing distributions slightly vary in the first few layers, but quickly become indistinguishable from layer to layer. One exception is in Switch routing where Math expert becomes dominant across tasks in the last model layer.

We observe that Code expert is a dominant force in Code domain in Top-2 routing with load balancing. Note the difference with other models where load balancing is not added, and Math expert prevails across domains. We look at Code domain closer and compare routing probability distribution for models with and without load balancing in Figure 5. On the bottom three graphs of the picture we can observe a phenomena of the dead expert, where routing probability to Code expert shifted to 0, while with load balancing added, probability distributions across experts look more similar, with slightly higher expectations for the Code expert.

To understand if experts specialize in other domains, we look closer at per-task distribution. Routing decision of the tokens in Math and Reasoning domains are shown in Figure 6. We observe that GSM8K task prefers Code and LLAMA-2 experts, while Math task more relies on in-domain expert. We hypothesise that this happens because GSM8K dataset consists of grade school math word problems that require common sense knowledge and basic arithmetic operations, while Math task requires college-level math knowledge, and more aligned with Math expert's training data. In the Reasoning domain, all tasks exhibit similar behaviour and equally rely on Math and generalist LLM's expertise.

|  | GSM8K | Human Eval | Natural Questions | ARC Challenge | MMLU | Average Score |
|---|---|---|---|---|---|---|
| BTX | 29.8 | 27.4 | 23.0 | 43.4 | 50.0 | 34.7 |
| no load-balancing (LB) | 34.6 | 19.5 | 23.2 | 44.4 | 51.6 | 34.6 |
| no LB & freeze experts | 34.8 | 18.3 | 24.1 | 44.9 | 51.4 | 34.7 |

Table 7: Ablations on different BTX training strategies. All variants are initialized from the same experts and trained for a total of 10B tokens during MoE finetuning.

| Task | Question and generation |
|---|---|
| GSM8K | Q: Janet's ducks lay 16 eggs per day. She eats three for breakfast every morning and bakes muffins for her friends every day with four. She sells the remainder at the farmers' market daily for $2 per fresh duck egg. How much in dollars does she make every day at the farmers' market? 
 A: Janet's ducks lay 16 eggs per day. She eats three for breakfast every morning. So she has 16 - 3 = 13 eggs left. She bakes muffins for her friends every day with 4. So she has 13 - 4 = 9 eggs left. She sells the remainder at the farmers' market daily for $2 per fresh duck egg. So she has 9 * 2 = 18 dollars. The answer is 18. |
| Human Eval | Q: from typing import List 

 def has_close_elements(numbers: List[float], threshold: float)->¿ bool: 
 """ Check if in given list of numbers, are any two numbers closer to each other than given threshold. 
 ¿ ¿ ¿ has_close_elements([1.0, 2.0, 3.0], 0.5) 
 False 
 ¿ ¿ ¿ has_close_elements([1.0, 2.8, 3.0, 4.0, 5.0, 2.0], 0.3) 
 True 
 """ 
 A: for i in range(len(numbers) - 1): 
 for j in range(i + 1, len(numbers)): 
 if abs(numbers[i] - numbers[j]) ¡ threshold: 
 return True 
 return False |
| Natural Questions | Q: who got the first nobel prize in physics 
 A: Max Planck |

Table 8: Examples of the token routing decisions for the Top-2 routing with load balancing in the math (GSM8K), code (Human Eval), and knowledge (Natural Questions) domains. Tokens highlighted are routed to the following experts: Wikipedia and LLAMA-2 7B, Math and LLAMA-2 7B, Code and LLAMA-2 7B, Math and Code, Wikipedia and Math, Wikipedia and Code. Tokens that were routed to the in-domain expert are underlined.

# D TFLOPs Comparison

The number of Floating-point operations (FLOPs) is often used for quantifying the efficiency of an algorithm. According to our approximate calculation (the active parameter count multiplied by the total number of training tokens), Sparse Upcycling (SU) in the compute-matched setting has 1.67E+10 TFLOPs while BTX has 2.39E+10. While the SU uses less FLOPs, looking only at FLOPs is flawed as it does not include communication cost. The main motivation of our work is to reduce the communication cost by making the training embarrassingly parallel. This improved efficiency cannot be measured in FLOPs alone and should be looked at together along with throughput. Such a comparison scheme has also been suggested in Dehghani et al. (2022) . Indeed, BTX has higher throughput (see Table 6), i.e. given the same amount of compute budget, BTX can do more computations.

# E LoRA Experts Training

Low-Rank Adaptation (LoRA) is a popular technique for making further training efficient for neural networks. We looked into only training a LoRA adaptor for each expert, but early experiments show that LoRA experts do not work well in our pretraining setting, possibly because LoRA experts do not have enough capacity for when experts are trained on and are required to fit massive amounts of data. In Table 9, we compared normal full-rank

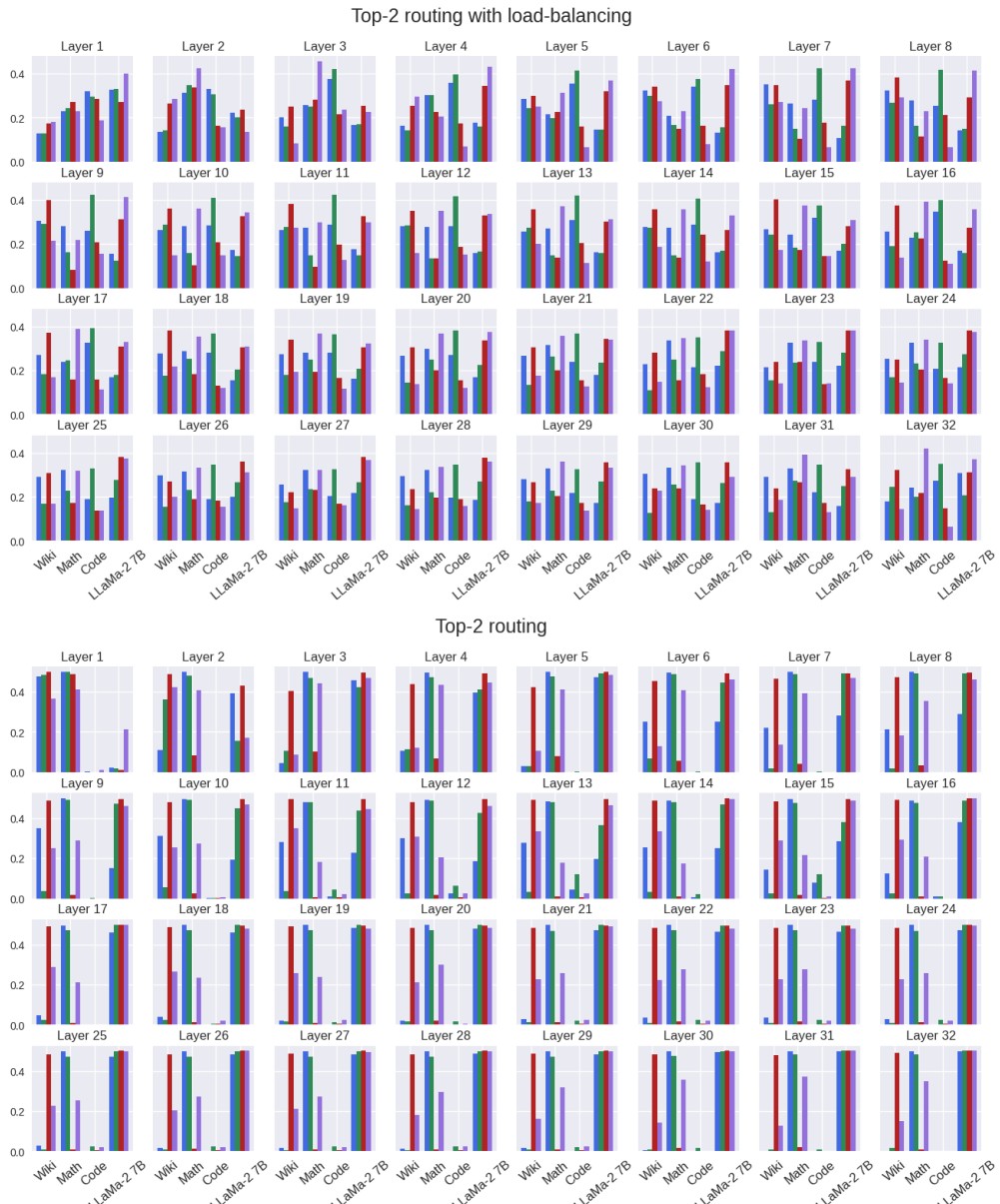

training against low-rank training (LoRA added to both attention and FFN parameters) for the code expert LLM training. We found that with rank=64, LoRA's performance lags the full-rank training as early as training for 40B tokens. We further found that training FFN with full-rank updates (while only adding LoRA on attention) can bridge that gap at small-data regime (e.g. 40B tokens) but LoRA still lags behind full-rank training when scaling up the training data to 200B tokens. This indicates that LoRA is unsuitable for a pre-training setup with large amounts of data.

## F  Limitations

Although our experimental results on BTX are promising, we have not fully explored its potential in this paper. Due to compute limitations, we only experimented with three domains and four experts in this paper. More domain experts can be added if data for more domains is available or using unsupervised domain discovery (Gururangan et al.,

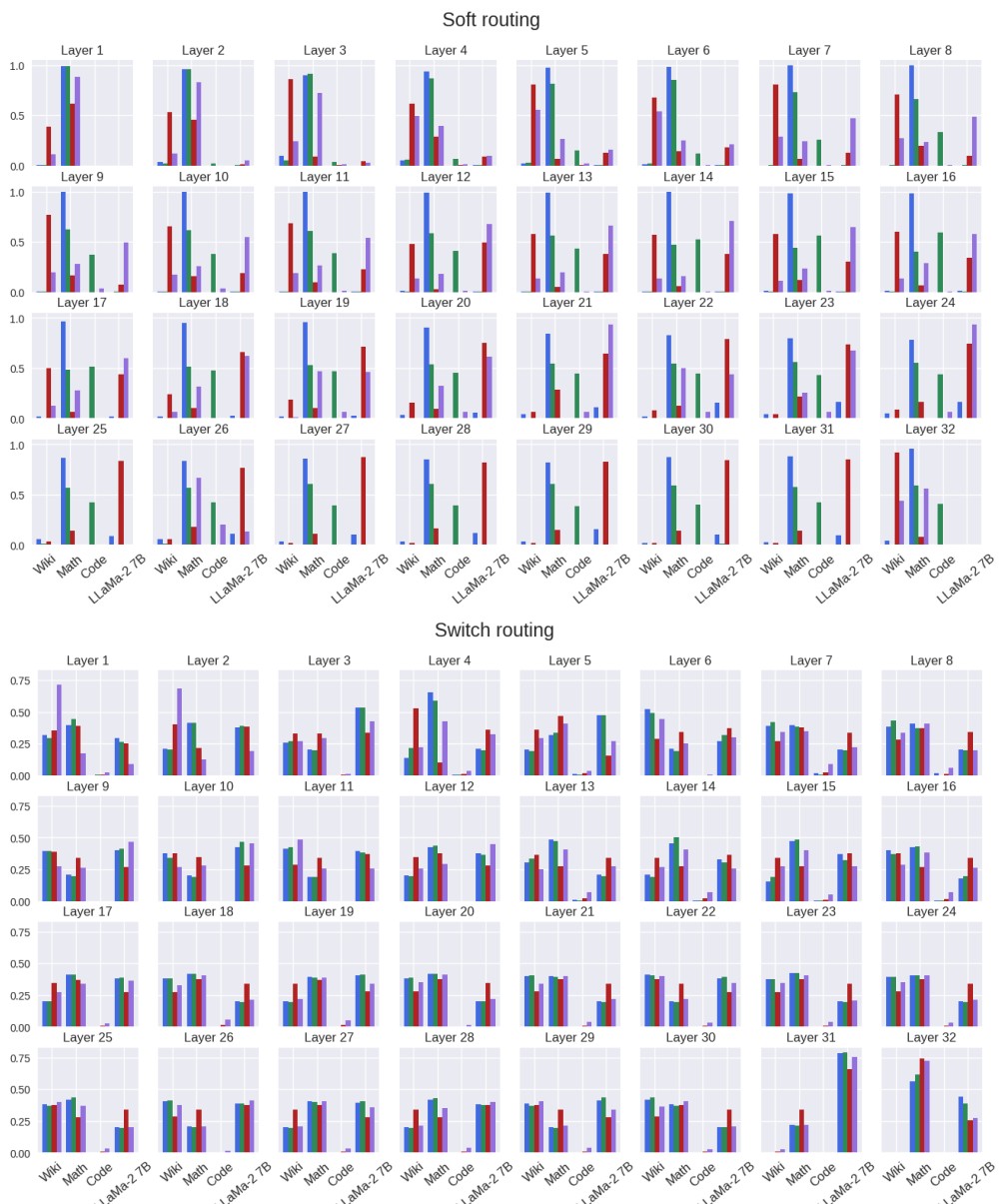

Figure 4: BTX routing decisions of the tokens at various layers to different experts (Wiki, Math, Code, LLAMA-2 7B) for different downstream tasks. The tasks are aggregated by domain: Code (Human Eval, MBPP), Math (GSM8K, MATH), World knowledge (Natural Questions, TriviaQA), and Reasoning (ARC-Easy, ARC-Challenge, SIQA, PIQA, and WinoGrande). We observe that top-2 routing with load balancing ensures more uniform distribution of the load between experts compared to the other routing methods across all layers.

2023). Then we can repeat the finetuning step using these new experts in addition to the previously trained experts. The MoE training itself should scale well to more experts as demonstrated in Fedus et al. (2022). Training on more domains should amplify the benefit of the parallelization of experts training. Having more experts will also make the final MoE model more efficient because the number of active experts can remain the same while its overall capacity increases.

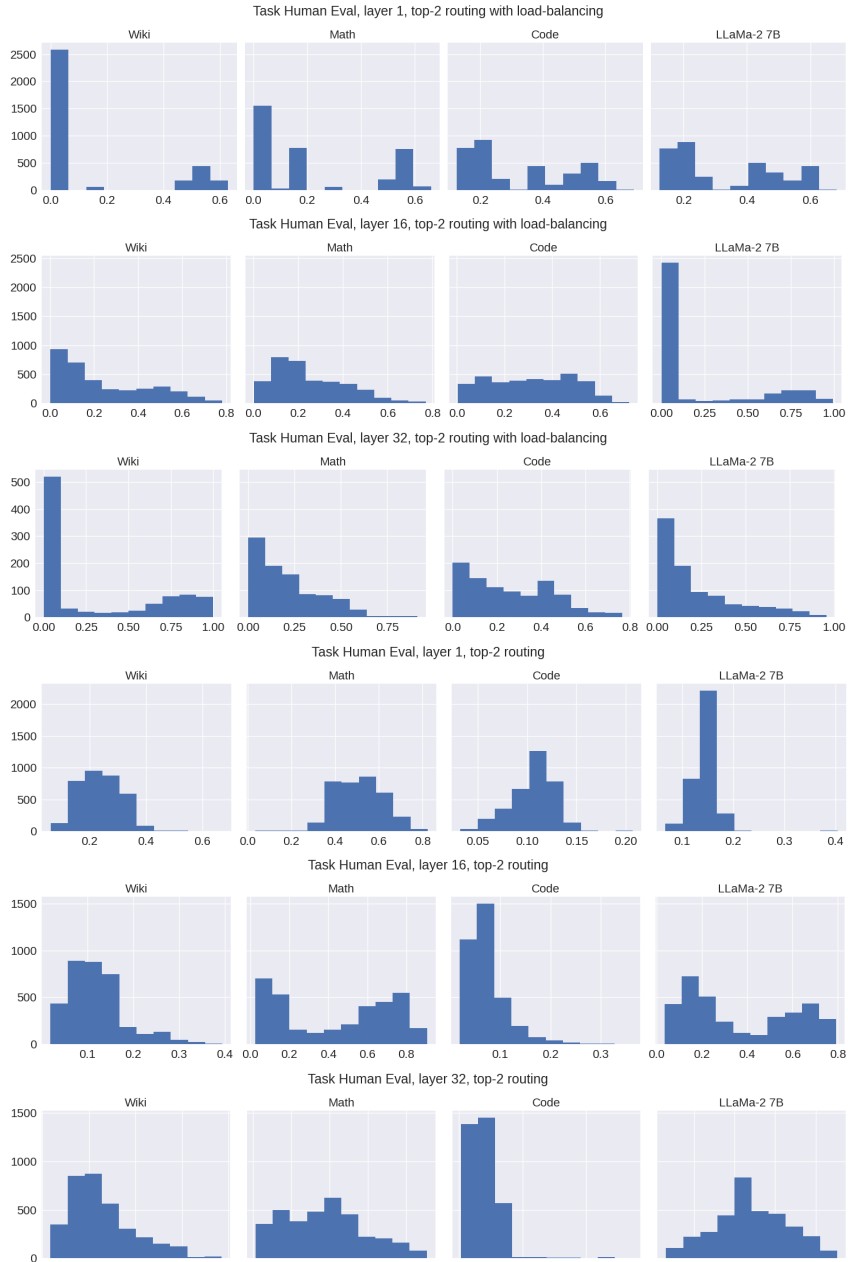

Figure 5: Routing probabilities per expert across different layers for Human Eval task. We compare top-2 routing with (left) and without load balancing (right).

In our experiments, we used a simple implementation of MoE and did not optimize it using more complex techniques such as placing different experts on different GPUs to run them in parallel. Such an efficient MoE implementation could shorten the training time of BTX, and the sparse upcycling baseline as well.

Compared to BTM, BTX provides an approach to finetune the combined experts, which can be directly applied in instruction finetuning or RLHF procedures. However, we leave that for future work as we focused on the pretraining stage in this paper.

The question of whether experts in MoE are better off specializing in specific domains or not is an interesting one that is worth further investigation. Our approach explicitly

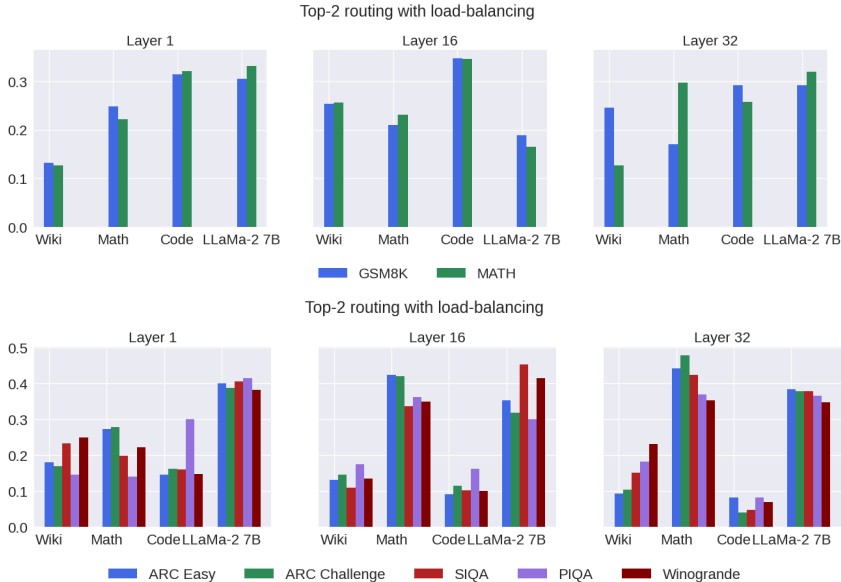

Figure 6: Routing decision of the tokens in Math and Reasoning domains. We observe that GSM8K task prefers Code and LLAMA-2 experts, while MATH task relies more on in-domain expert. In the Reasoning domain, the load is distributed between Math and LLAMA-2 7B experts.

| #tokens | Training | Valid ppl | HumanEval (pass@1) |
|---------|----------|-----------|--------------------|
| 40B | Full-rank | 1.86 | 21.34 |
| 40B | LoRA (attn r=64, ffn r=64) | 1.99 | 15.24 |
| 40B | LoRA (attn r=64, ffn full-rank) | 1.88 | 24.39 |
| 200B | Full-rank | 1.74 | 31.71 |
| 200B | LoRA (attn r=64, ffn full-rank) | 1.77 | 28.66 |

Table 9: Ablations on parameter-efficient experts with LoRA.

tied experts to certain domains, but such specialization does not seem to emerge naturally during MoE training (Jiang et al., 2024). We observed that some experts are used more in their corresponding domain tasks, showing that their domain specialization partially remains even after the MoE finetuning.

We only compared BTX to two of its special variants, i.e. BTM with 100% compute allocated to expert training and 0% on MoE finetuning, and sparse upcycling with 0% compute allocated to expert training and 100% on MoE finetuning. Future work could perform a thorough sweep of the compute allocation ratio between expert training and MoE training. Also, we did not perform experiments with different data mixtures for MoE finetuning other than uniform sampling.

