# OpenReview forum: "Branch-Train-MiX: Mixing Expert LLMs into a Mixture-of-Experts LLM"
_colmweb.org/COLM/2024/Conference — COLM_

### Official Review · Reviewer_go34 · 2024-05-08

**Rating:** 7
**Confidence:** 4
**Ethics Flag:** 1

**Summary:**

The paper introduces Branch-Train-MiX (BTX), a method for training Large Language Models (LLMs) with expertise in multiple domains. BTX branches a seed model to train experts in parallel, then integrates them into a Mixture-of-Experts (MoE) model and fine-tunes for token-level routing. This approach aims to enhance the seed model's performance across various tasks while maintaining efficiency.

**Reasons To Accept:**

- BTX's parallel and asynchronous training of experts reduces communication costs and increases throughput.
- It demonstrates better compute-performance trade-offs compared to training a larger generalist LLM or multiple specialized ones.

**Reasons To Reject:**

- The authors present only three domains and four experts, making the results less convincing. Admittedly, full model training is expensive, I suggest the authors report results with PEFT techniques on more domains and experts.

---

> ### Author Rebuttal · Authors · 2024-05-31
>
> Thank you for reviewing our paper and providing insightful feedback.
>
> > The authors present only three domains and four experts, making the results less convincing.
>
> We agree with the reviewer that further evaluation of the proposed model would be beneficial to strengthen the conclusions of our work, however due to compute and resource limitations, we were unable to include more experts. We tested our approach on the most common tasks - mathematical and code reasoning, and commonsense knowledge. However, even with only 4 experts, we see the benefit of having multiple experts from the large improvements compared to a dense baseline.
>
> > Admittedly, full model training is expensive, I suggest the authors report results with PEFT techniques on more domains and experts.
>
> We looked into PEFT using LoRA experts, but from our experience we found that LoRA experts don't work well in the pretraining setting possibly because LoRA experts do not have enough capacity when experts are trained on massive amounts of data.

---

> > ### Comment · Reviewer_go34 · 2024-06-05
> >
> > Thank you for your response.
> >
> > Regarding the experiments that I found less convincing, I still have some concerns.
> >
> > I appreciate the insights on the LoRA experts during pretraining. I've learned a lot from this discussion. I hope that the results of these seemingly unsuccessful experiments can be included in the appendix. **I will raise my rating** if you can provide this information (which will help more practitioners save unnecessary training costs).

---

> > > ### Author Response · Authors · 2024-06-06
> > >
> > > We agree that unsuccessful experiments can be useful, and we will add our LoRA results in the appendix.
> > >
> > > Below is our experimental result using LoRA. We compared normal full-rank training against low-rank training (LoRA added to both attention and FFN parameters) for the code expert LLM training. We found that with rank=64, LoRA’s performance lags behind the full-rank training as early as training for 40B tokens. We further found that training FFN with full-rank updates (while only adding LoRA on attention) can bridge that gap at small-data regime (e.g. 40B tokens) but LoRA still lags behind full-rank training when scaling up the training data to 200B tokens. This indicates that LoRA is unsuitable for pre-training setups with large amounts of data.
> > >
> > > #tokens | Training ________ | Valid ppl | HumanEval (pass@1)
> > >
> > > 40B ____ | Full-rank _______ | 1.86 |  21.34
> > >
> > > 40B ____ | LoRa___________ | 1.99 |  15.24
> > >
> > > 40B ____ | LoRa (attn only) | 1.88 |  24.39
> > >
> > > 200B___ | Full-rank  _______ | 1.74 |  31.71
> > >
> > > 200B___ | LoRa (attn only) | 1.77 |  28.66

---

### Official Review · Reviewer_tFfH · 2024-05-10

**Rating:** 6
**Confidence:** 4
**Ethics Flag:** 1

**Summary:**

This paper introduces the paradigm of aggregating multiple dense models finetuned on different data into an intact MoE, which first separately finetunes $N$ domain LLMs with BTM, then aggregates these models into a MoE through further finetuning. It also proposes various variants with different choices of the gating networks and initializations of expert parameters, and conducts detailed experiments. The results show salient improvements of BTX on the downstream performance under Data Matching (DM) and Compute Matching (CM) settings.

**Questions To Authors:**

Please see the reasons to reject.

**Reasons To Accept:**

1. The idea of merging finetuned LLMs to MoE is intuitive and interesting.
2. The introduced pipeline seems acceptable considering the trade-off between the performance and efficiency.
3. Results are good under either the same amount of data (DM) or the same computation (CM).

**Reasons To Reject:**

1. **The comparison to the upcycling baseline is not fair.**
   - As far as I know, the compute budget of MoE highly depends on the library and hardwire environment used. So the Compute Matching (CM) may not be a fair comparison as MoE would encounter more communication bottlenecks as you have shown in Table 6, which could be enhanced by more painless designs of training frameworks for MoE.
   - Also, the Data Matching (DM) is not fair for the Upcycling as its total parameter activation is less than BTX. BTX activates all the parameters in the first stage and switches to top-$k$ in the second finetuning stage. However, the upcycled MoE always uses top-$k$ activation.
   - A fair comparison should be under the same TFLOPs which is not included.

2. **Lack of comparison to the individual domain expert LLMs.**
   - In Table 2, the specialized LLMs should include the individual domain expert LLMs in Table 1 for clarification of the gap between the general LLM and domain oracle LLMs. We are curious about whether the combined model can surpass the oracle model in the specified domains, rather than other standalone LLMs.

---

> ### Author Rebuttal · Authors · 2024-05-31
>
> > the compute budget of MoE highly depends on the library…
>
> MoE performance can vary depending on the implementation, so it is possible to increase the efficiency of MoE training using more complex training schemes. In contrast, the first step of BTX offers a simple recipe for training a large model while being less dependent on complex training infrastructures and implementations while being more robust to hardware failures. However, we still think MoE training is a very effective way of scaling, and we showed that it can be effectively used as the final step in BTX. This also means that an efficient MoE implementation will also make BTX more efficient, but of course to a lesser degree compared to full MoE training.
>
> > the Data Matching (DM) is not fair … BTX activates all the parameters in the first stage
>
> In the data matching case, both models see the same amount of data during training. However, BTX does not activate all parameters on every data point in the first stage. For example, if the sample is from the math dataset, then only the math expert LLM will process this data. Therefore, only 1/4 of the total BTX parameters will be activated during the first stage of training. In contrast, sparse-upcycling activates 2 out of 4 FF experts.
>
> > A fair comparison should be under the same TFLOPs
>
> Sparse upcycling (SU) in the compute-matched setting has 1.67E+22 FLOPS while BTX is 2.39E+22. The SU uses less FLOPS, but looking only at FLOPS is flawed as it doesn’t include communication cost. The main motivation of BTX was to reduce the communication cost by making the training embarrassingly parallel. This improved efficiency cannot be measured in FLOPS alone but should be looked at together along with throughput. Such a comparison scheme has also been suggested in https://arxiv.org/abs/2110.12894 . Indeed, BTX has higher throughput (926 GPU days vs SU’s 1007), i.e. given the same amount of compute budget, BTX can do more computations (higher TFLOPs). We will include FLOPS and discussion in the paper.
>
> > In Table 2, the specialized LLMs should include the individual domain expert LLMs
>
> We will add a table comparing BTX vs.single-expert on tasks in each domain, but cannot show it here due to the space limit. In the math and code domains, the final model performance is little behind the expert LLMs, but overall it has balanced good performance. We will include this table in the paper. We can reply to you if you want to see the full table.

---

> > ### Comment · Reviewer_tFfH · 2024-06-04
> >
> > Thanks for your response.

---

### Official Review · Reviewer_EfwL · 2024-05-15

**Rating:** 8
**Confidence:** 5
**Ethics Flag:** 1

**Summary:**

In this paper, the authors propose a new method to train domain expert LLMs and mix them together to get a better model on all the tasks. Compared to the existing studies, it has two major differences - 1) compared to sparse upcycling (SU), it has a separate domain expert training stage which each individual expert trained on special tasks. 2) compared to branch-train-merge (BTM), combining different expertise is done as a single model rather than ensemble using MoE architecture. The authors show that the proposed method can outperform existing SU and BTM on average scores for various tasks. The authors also show the proposed method can be more efficient than other method thanks to the parallel training of different domains.

Overall, the paper is very well-organized and easy to follow. The proposed idea looks interesting and intuitive, even though the gains in the quality are not huge (compared llama2-13b and SU). However, it will be a good contribution to the community to improve more in that direction.

**Questions To Authors:**

- It will be nice to have a discussion how to extend this to the greater number of experts and larger models. Many open-source models use at least 8 experts.
- It will be also useful to have a baseline MoE model which has the same number of parameters (4 experts) but trained from scratch.

**Reasons To Accept:**

- Paper is very well written
- The idea is simple and intuitive, yet it gives some good gains in quality and efficiency.
- It has various ablation studies including routing analysis, compute efficiency and etc.

**Reasons To Reject:**

- I don't see a good reason to reject this paper.

---

> ### Author Rebuttal · Authors · 2024-05-31
>
> We would like to thank the reviewer for their time and careful consideration.
>
> > It will be nice to have a discussion how to extend this to the greater number of experts and larger models. Many open-source models use at least 8 experts.
>
> Extending to more experts should be relatively straightforward. First, one needs to identify more domains and collect data corresponding to those domains, then train corresponding experts starting from the seed model. The previously trained expert LLMs can be used as is. The mixing and fine-tuning steps can be applied to 8 experts or more in the same way. As you mentioned, there is evidence that MoE routing scales well to more experts. We expect this to be true in our MoE fine-tuning step as well. We will add this discussion to the paper.
>
> > It will be also useful to have a baseline MoE model which has the same number of parameters (4 experts) but trained from scratch.
>
> Yes, it would be interesting to compare against such a baseline. However, in our setup, training any model from scratch will take too much computation, especially for a larger MoE model. Still, there is evidence that sparse-upcycling works well, even compared to MoE trained from scratch as shown in [1] (see their Fig 4). We quote from [1] “The language MoE model trained from scratch requires about 120% of the original dense checkpoint’s computation budget to catch up to the upcycled model”. Also, prior work on BTM [2] compared with training MoE from scratch (although with domain dependent routing) and they found the asynchronous expert training is more compute efficient. And in our work, we show that BTX is superior to BTM.
>
> [1] Komatsuzaki, Aran, et al. "Sparse upcycling: Training mixture-of-experts from dense checkpoints." (2022) https://arxiv.org/pdf/2212.05055
>
> [2] Li, Margaret, et al. "Branch-train-merge: Embarrassingly parallel training of expert language models." (2022) https://arxiv.org/pdf/2208.03306

---

> > ### Comment · Reviewer_EfwL · 2024-06-04
> >
> > Thanks for the rebuttal.
> >
> > I agree on your points, and keep my score.

---

### Decision · Program_Chairs · 2024-07-10

**Decision:**

Accept

**Comment:**

This work proposes a new way to train a mixture of expert model called Branch-Train-MiX (BTX). It starts with parallel finetuning of LLMs to train experts for different tasks. It then merges those experts into a single model as follows. The FeedForward layer of each model is combined together to create a single MoE layer for the end model. A router routes each token to select expert feedforward models. The self-attention layer is merged by averaging the weights across models. The end model is the finetuned on the aggregated data to train a generalist router. The experts are trained on math, coding, and general text (wikipedia). The method shows superior performance on several downstream tasks (including math, coding, knowledge-based questions, and reasoning) over prior approaches such as sparse upcycling and Branch-Train-Merge approaches.

Overall, the independent and parallel finetuning of experts and then merging them using an MoE concept is an interesting and scalable way to train a generalist agent and provides a path for maintaining the benefits of finetuning while minimizing its adverse affects on the general capabilities of the model. This line of work can ignite new ideas on reusing and building on top of finetuned models in efficient ways towards a generalist model.